# Fasting and Exercise Induce Changes in Serum Vitamin D Metabolites in Healthy Men

**DOI:** 10.3390/nu13061963

**Published:** 2021-06-08

**Authors:** Małgorzata Żychowska, Rafał Rola, Andżelika Borkowska, Maja Tomczyk, Jakub Kortas, Katarzyna Anczykowska, Karol Pilis, Konrad Kowalski, Wanda Pilch, Jędrzej Antosiewicz

**Affiliations:** 1Department of Sport, Faculty of Physical Education, Kazimierz Wielki University, 85-091 Bydgoszcz, Poland; malgorzata.zychowska@ukw.edu.pl; 2Masdiag Sp. z o.o. Company, 01-882 Warsaw, Poland; rafal.rola@masdiag.pl (R.R.); konrad.kowalski@masdiag.pl (K.K.); 3Faculty of Chemistry, Nicolaus Copernicus University, 87-100 Torun, Poland; 4Department of Bioenergetics and Physiology of Exercise, Medical University of Gdansk, 80-210 Gdansk, Poland; andzelika.borkowska@gumed.edu.pl; 5Department of Bioenergetics and Nutrition, Faculty of Physical Education, Gdansk University of Physical Education and Sport, 80-336 Gdansk, Poland; tomczykmaja@gmail.com; 6Department of Health and Life Sciences, Gdansk University of Physical Education and Sport, 80-336 Gdansk, Poland; jakub.kortas@awf.gda.pl; 7Department of Biochemistry, Faculty of Physical Education, Gdansk University of Physical Education and Sport, 80-336 Gdansk, Poland; kasia.anczykowska@gmail.com; 8Department of Health Sciences, Jan Dlugosz University, 42-200 Czestochowa, Poland; k.pilis@ujd.edu.pl; 9Institute for Basics Sciences, Faculty of Physiotherapy, University of Physical Education in Krakow, 31-571 Krakow, Poland; wanda.pilch@awf.krakow.pl

**Keywords:** food restriction, endurance exercise, 1,25(OH)_2_D_3_, 3-*epi*-25(OH)D_3_, 24,25(OH)_2_D_3_, 25(OH)D_3_

## Abstract

Background: Vitamin D plays pleiotropic roles in the body and hence, changes in its metabolism and distribution during starvation could play an important role in the adaptive response to famine. We aimed to identify the responses of some vitamin D metabolites to 8 d of fasting and exercise. Methods: A repeated-measures design was implemented, in which 14 male volunteers fasted for 8 d and performed an exercise test before and after fasting. Serum samples were collected on day 1 after night fasting and after 8 d of complete food restriction, before and 1 h and 3 h after exercise. Results: After 8 d of fasting, compared with baseline values, serum 24,25(OH)_2_D_3_ and 3-*epi*-25(OH)D_3_ levels significantly increased; those of 25(OH)D_3_ and 1,25(OH)_2_D_3_ were unaffected; and those of 25(OH)D_2_ decreased. Exercise on the first day of fasting induced an increase in serum 3-*epi*-25(OH)D_3_ levels, while exercise performed after 8 d of fasting induced an increase in 25(OH)D_3_, 24,25(OH)_2_D_3_, 25(OH)D_2_, and 3-*epi*-25(OH)D_3_ levels. Conclusion: Increases in 24,25(OH)_2_D_3_ and 3-*epi*-25(OH)D_3_ levels imply that fasting stimulates vitamin D metabolism. The effects of exercise on serum vitamin D metabolites, which are most pronounced after fasting and in subjects with serum 25(OH)D_3_ above 25 ng/mL, support the notion that fasting and exercise augment vitamin D metabolism.

## 1. Introduction

Adaptation of the human body to starvation is crucial for survival and well-being during periods of famine. Short fasting (i.e., a few days) is popular among religious people, and those who care about their health and body mass, etc. However, the mechanism of the adaptive response of the human body to starvation is not fully understood. 

Fat tissue plays an important role during fasting as a source of energy. However, it also stores many fat-soluble vitamins, including vitamin D. Vitamin D directly and indirectly regulates multiple processes in the human body. Its molecular mechanism of action is related to the activation of vitamin D receptor, which is present in the majority of tissues in humans [1,2]. The active form of vitamin D is 1,25(OH)_2_D_3_. It activates the vitamin D receptor and thus regulates the expression of approximately 900 genes [3]. Hence, adequate vitamin D status is indispensable for proper function of the human body and mind. 

Limited exposure of the skin to ultraviolet radiation emitted by the sun and low consumption are considered to be the main reasons for vitamin D deficiency. Skin-synthesized or food-ingested vitamin D undergoes activation by CYP2R1 and CYP27B1 to form 25(OH)D_3_ and 1,25(OH)_2_D_3_, respectively. Some other metabolites of vitamin D are also formed, e.g., 3-*epi*-25(OH)D_3_ and 24,25(OH)_2_D_3_, which are considered to be inactive forms of vitamin D. However, recent studies indicate that they do have some biological activity, such as stimulation of the antioxidant potential, reduction of the serum parathormone levels, and others [4,5]. In addition, 24,25(OH)_2_D_3_ is a part of a signaling loop, as 1,25(OH)_2_D_3_ stimulates its formation and 24,25(OH)_2_D_3_ protects against 1,25(OH)_2_D_3_ toxicity. Several studies have demonstrated that vitamin D is a potent anti-inflammatory agent. For example, it has been shown that it can inhibit biosynthesis of proinflammatory cytokines and can upregulate anti-inflammatory cytokines [6,7]. Furthermore, elevated levels of 3-*epi*-25(OH)D_3_ and 24,25(OH)_2_D_3_ are associated with an improved cardiovascular risk profile and a reduced risk of death among patients with chronic kidney disease [8]. Serum 25(OH)D_3_ levels are considered to be a marker of vitamin D status. 

Sun exposure and food intake are not the only factors influencing vitamin D status. Adipose tissue and skeletal muscle can accumulate and store vitamin D. Further, an increase in the body fat mass is associated with a blunted rise in 25(OH)D_3_ levels following supplementation, which is possibly caused by vitamin D accumulation in that tissue. In addition, while skeletal muscle actively accumulates vitamin D, no association between skeletal muscle mass and serum 25(OH)_2_D_3_ has been reported [9]. Furthermore, exercise can increase serum 25(OH)D_3_ levels, possibly by stimulating its release from adipose tissue and skeletal muscle [10], and modulating its metabolism. 

Long-term food restriction leads to weight loss, especially that of adipose tissue. Hence, one could anticipate that it would also lead to major vitamin D redistribution into the blood and possible changes in its metabolism; however, the data are scarce. In the current study, we measured the levels of four major metabolites of vitamin D in individuals who had starved themselves for 8 d. We then examined the associations of vitamin D metabolites with changes in body composition and markers of inflammation. We showed that fasting stimulates vitamin D metabolism and modulates the effects of exercise on this metabolism. These data imply that vitamin D metabolites have some important role in adaptation to food shortage.

## 2. Materials and Methods

### 2.1. Ethics

Study participants had already practiced fasting prior to the study, and the current study was designed to fit in with their ensuing fasting session. The participants remained under medical supervision during the study period and were informed by medical personnel about the possible negative health effects of such practice. For safety reasons, the men remained under medical care for 3 days before the study, during the 8 days of fasting, and 3 days after its completion.

All participants were informed about the purpose and methods of the study, and gave their written consent prior to the study.

The current study was approved by the Committee for Ethics in Scientific Research of Jan Dlugosz University in Czestochowa (Poland; KE-0/1/2019; 5 March 2019). It was performed in compliance with the principles of the Declaration of Helsinki—Ethical Principles for Medical Research Involving Human Subjects.

### 2.2. Study Group Characteristics 

Fourteen healthy men participated in the study (Figure 1). The number of fasts performed by the respondents before the study ranged from 3 to 12, with the last one at least 6 months prior to the study. All subjects who qualified for the study underwent a medical examination and no contradictions to performing exercise were found. The participants declared that they had performed moderate-intensity physical activity in the form of yoga. The authors of the study did not control the physical fitness levels of individual participants. The study did not include a control group as the effects of exercise had been compared before and after fasting in the same subject, thus they were a control for themselves. 

The inclusion criteria were as follows: experience in performing fasting of more than 3 d; age between 30 and 70 years; body weight between 60 and 100 kg; body mass index (BMI) range of 20–29.9 kg/m^2^; no chronic diseases; systolic blood pressure in the range 100–140 mmHg; and diastolic blood pressure in the range 60–90 mmHg. The exclusion criteria were as follows: smoking cigarettes, the use of medications, any diseases; strong stimulants, or psychoactive substances; and failure to complete the test procedure. Table 1 summarizes the basic somatic parameters of the participants. One subject was excluded from the analysis as he used vitamin D supplements (around 10,000 IU per day) for several weeks before the study. 

During the fasting period, the subjects only consumed mineral water, which contained an average amount of ions. No food or drink containing calories were allowed. During the study no adverse effects of fasting were recorded. 

### 2.3. Anthropometric Measurements

The study participants’ age and basic somatic data were recorded (body height, body weight, body fat content, fat-free mass (FFM), total body water (TBW), and BMI). The measurements of somatic body parameters were performed in the morning using the bioelectric impedance method (TBF 300A body composition analyzer, Tanita, Amsterdam, The Netherlands) after a 12-h fast and a night’s rest, having abstained from alcohol, medication, and exercise for the previous 2 d.

### 2.4. Exercise Test

The study participants performed an exercise test with increasing intensity until exhaustion using an Excalibur Sport cycloergometer (Lode B.V., Groningen, The Netherlands). The initial load (60 W) was increased gradually every 3 min by 30 W. The test was interrupted when the subject’s oxygen uptake began to stabilize at the maximum level or decreased; when the subject was unable to maintain the rhythm of pedaling; or when the heart rate did not increase, or stabilized at its maximum level or started to decrease. All procedures described above were performed before and after 8 d of complete fasting. 

### 2.5. General Blood Parameter 

Ten milliliters of venous blood were collected from study participants for biochemical analyses. To determine β-hydroxybutyrate level the blood was collected twice: before the first exercise (baseline) and after 8 d of starvation (before the second exercise test). β-hydroxybutyrate levels were determined using a commercial kit (RANBUT diagnostic kit, RANDOX, Crumlin, UK). Uric acid levels were determined using an enzymatic reaction (ABBOTT’s ARCHITECT c SYSTEM 4000 analyzer, Abbott Park, IL, USA). Plasma IL-10 was determined via an enzyme immunoassay method using commercial kits (EIAAB, Wuhan EIAab Science Co., Ltd., Wuhan, China, Catalog No: E0056h). The intra-assay coefficients of variability (CVs) and inter-assay CVs reported by the manufacturer were ≤4.7% and ≤6.2%, respectively.

### 2.6. Measurements of Serum Vitamin D Metabolites 

Venous blood samples for vitamin D metabolites were collected at three time points (1 h before, and 1 h and 3 h after the exercise) into tubes containing a coagulation accelerator. The serum was separated using a laboratory centrifuge, aliquotted into 1.5-mL centrifuge tubes, and frozen at −80 °C until further analysis. 

Before the analysis, serum proteins were precipitated and derivatized using a Cookson-type reagent (DAPTAD). Proteins were precipitated using acetonitrile. For 1,25(OH)_2_D_3_ determination, the sample preparation involved liquid–liquid extraction with ethyl acetate. Quantitative analyses were performed using liquid chromatography coupled with tandem mass spectrometry (Exion LC system coupled with QTRAP4500, Sciex, Framingham, MA, USA). Chromatographic separation was carried out using an XDB-C18 column (50 × 4.6 mm, 1.7 μm; Agilent, Santa Clara, CA, USA). Serum samples were analyzed in the positive ion mode, using electrospray ionization. The concentrations of the following vitamin D metabolites were determined: 25(OH)D_3_, 24,25(OH)_2_D_3_, 1,25(OH)_2_D_3_, 3-*epi*-25(OH)D_3_, and 25(OH)D_2_. The concentration range was 1–100 ng/mL for 25(OH)D_3_; 0.1–10 ng/mL for 25(OH)D_2_, 3-*epi*-25(OH)D_3_, and 24,25(OH)_2_D_3_; and 10–200 pg/mL for 1,25(OH)_2_D_3_. In addition, the ratios of 25(OH)D_3_ to 24,25(OH)_2_D_3_, and 25(OH)D_3_ to *epi*-25(OH)D_3_ were calculated.

### 2.7. Statistical Analysis

Data are given as means ± SD. Statistical analyses were performed by using Statistica 13.1 software. For normal distribution results, an unpaired *t*-test analysis was performed to identify significant differences at baseline. For the remaining results, Mann–Whitney testing was used. Analyses of changes in vitamin D metabolite levels induced by 8 d of fasting were performed: for normally distributed variables, a paired *t*-test was used; and for others, a Wilcoxon test was used. Similar analyses were performed for changes after one exercise, separately for baseline and after fasting. Then, 2 separate (group: 25(OH)D_3_ < 20 ng/mL, 25(OH)D_3_ > 20 ng/mL) × 2 (time: PRE, POST) repeated measures analyses of variances (rANOVA) were calculated. In case of a significant time × group interaction, for homogenous results, Tukey’s post-hoc test for unequal sample sizes was performed to identify significant differences. For heterogeneous results, an ANOVA Friedman’s test and a Dunn–Bonferroni post-hoc test were used. The effect size (partial eta squared, $\eta_{p}^{2}$) was also calculated, with $\eta_{p}^{2}$ ≥ 0.01 indicating a small effect, ≥0.059 indicating a medium effect, and ≥0.138 indicating a large effect. The relationships between variables were evaluated using the Spearman correlation coefficient. The level of significance was set at *p* < 0.05.

## 3. Results

### 3.1. Effect of 8 d of Fasting on Body Composition and Performance

Fifteen men (aged 57 ± 14 years, min. 27, max. 64 years) volunteered to participate in the study. One individual was excluded because of high 25(OH)D_3_ levels before the experiment (65.9 ng/mL). After 8 d of fasting, statistically significant increases in β-hydroxybutyrate (0.3 ± 0.2 mM vs 4.6 ± 1.1 *p* < 0.01) and changes in body composition and performance levels were observed (Table 1). 

### 3.2. General Outcomes

Eight days of fasting induced statistically significant changes in the levels of three vitamin D metabolites: 24,25(OH)_2_D_3_ (1.5 ± 1.0 to 1.7 ± 1.2 ng/mL, *p* = 0.04), 3-*epi*-25(OH)D_3_ (1.0 ± 0.3 to 1.5 ± 0.6 ng/mL, *p* < 0.01), and 25(OH)D_2_ (1.1 ± 0.4 to 0.8 ± 0.3 ng/mL, *p* < 0.01) (Figure 2).

Statistically significant correlations between the levels of 25(OH)D_3_ and the following metabolites were observed at baseline: 24,25(OH)_2_D_3_ (r = 0.94, *p* < 0.01), 25(OH)D_2_ (r = 0.52, *p* = 0.05), 3-*epi*-25(OH)D_3_ (r = 0.85, *p* < 0.01), and 1,25(OH)_2_D_3_ (r = 0.87, *p* < 0.01). Similar correlations were observed after 8 d of fasting. 

Body weight loss was noted in all participants (mean change: −5.6 ± 1.4 kg, i.e., 7% reduction; Table 1). At the same time, 3-*epi*-25(OH)D_3_ levels increased in 13 participants (mean change: 0.5 ± 0.4 ng/mL) and slightly decreased in one individual (–0.1 ng/mL). These changes were positively correlated (r = 0.69, *p* = 0.01), which indicates that a decrease in body weight impacts the increase in 3-*epi*-25(OH)D_3_ levels.

In addition, a decrease in body weight was significantly correlated with a decrease in 25(OH)D_3_ levels (r = 0.71, *p* < 0.01). Therefore, it can be deduced that a 12% decrease in body weight would result in a statistically significant decrease in 25(OH)D_3_.

Finally, no statistically significant change in IL-10 levels (69.5 ± 37.0 to 74.0 ± 43.4 pg/mL, *p* = 0.66) was observed in response to fasting.

### 3.3. Effect of Fasting on Vitamin D Metabolite Response to a Single Exercise

An increase in vitamin D metabolite levels was observed following a single exercise regardless of fasting. However, the increase was statistically significant only after 8 d of fasting, with the exception of 3-*epi*-25(OH)D_3_ levels, which increased after exercise at baseline (Table 2). 

### 3.4. Subgroup Analysis

To evaluate if baseline 25(OH)D_3_ levels affect the impact of fasting and exercise on 25(OH)D_3_ metabolism, the participants were divided into two subgroups based on baseline 25(OH)D_3_ levels, assuming a cut-off of 20 ng/mL. There were eight individuals in the group with levels below 20 ng/mL, and six in the group with levels over 20 ng/mL.

Significant differences were noted in the remaining vitamin D metabolites with the exception of 25(OH)D_2_ (*p* = 0.33, Table 3).

Post-hoc testing revealed an increase in 3-*epi*-25(OH)D_3_ levels after 8 d of fasting and a significant decrease in 25(OH)D_2_ levels (*p* = 0.02) in the subgroup with higher baseline 25(OH)D_3_ values. In the other subgroup, a significant change (decrease) was noted only for 25(OH)D_2_ levels (*p* < 0.01, Figure 3).

No differences in changes in the IL-10 levels were apparent between the analyzed subgroups (*p* = 0.21, $\eta_{p}^{2}$ = 0.15).

Finally, a significant correlation between VO_2_ levels and vitamin D metabolite levels was observed (25(OH)D_3_: r = 0.56; *p* = 0.04; 24,25(OH)_2_D_3_: r = 0.62, *p* = 0.02; 25(OH)D_2_: r = 0.74, *p* = 0.02; and 3-*epi*-25(OH)D_3_: r = 0.64, *p* = 0.02).

Detailed analysis of the results according to the baseline 25(OH)D_3_ levels revealed a significantly different response to exercise within the study subgroups (Table 4). Increases in 25(OH)D_3_ and 24,25(OH)_2_D_3_ levels were observed only in the subgroup with 25(OH)D_3_ baseline levels over 20 ng/mL.

## 4. Discussion

In the current study, we asked whether fasting-induced body mass reduction would be associated with changes in vitamin D status and its metabolism. We demonstrated that 8 d of fasting by itself induced changes in vitamin D metabolism and modulated the effects of exercise on that metabolism. To the best of our knowledge, this is the first report in which the effect of fasting on vitamin D metabolism was evaluated in human subjects. 

Adipose tissue and skeletal muscle store substantial amounts of vitamin D [11]. Further, a rise in blood 25(OH)D_3_ levels after exercise has been attributed to increased lipolysis. Hence, it can be expected that during starvation, when lipolysis is activated, some of the stored vitamin D is released into the blood. We observed a significant reduction of the body mass, fat mass, and FFM after 8 d of fasting but, surprisingly, serum 25(OH)D_3_ levels did not change. According to studies involving animal models, 3 d of fasting increase serum 25(OH)D_3_ levels and subcutaneous adipose tissue cholecalciferol levels [12]. Further, 30-min of cycling exercise and ultramarathon run, which certainly stimulate lipolysis, increase serum 25(OH)D_3_ levels to a similar extent, even if the latter is accompanied by a significant reduction of body fat [10,13,14]. These observations imply that the change in serum 25(OH)D_3_ levels is a regulated process and not simply an effect of increased lipolysis in adipose tissue. It has been also reported that skeletal muscle might be the largest pool of 25(OH)D_3_ in the whole body and that it may supply vitamin D to the serum [15]. The mechanism of vitamin D transport into and from skeletal muscle is not fully understood. Hence, it cannot be excluded that during starvation, a portion of vitamin D stores is discharged from adipose tissue and enters the skeletal muscle. Lack of changes in serum 25(OH)D_3_ levels after starvation can also be a consequence of its increased metabolism. In the current study, levels of metabolites, such as 24,25(OH)_2_D_3_, 3-*epi*-25(OH)D_3_, and 1,25(OH)_2_D_3_, increased after 8 d of fasting, which can significantly influence the 25(OH)D_3_ status. Furthermore, in animal models, 12–24 h of fasting downregulate gene expression and protein levels of hepatic CYP2R1, an enzyme that catalyzes vitamin D 25-hydroxylation [16]. However, as this enzyme is present in the liver as well as in many other tissues, including skeletal muscle and adipose tissue, it is unlikely that the formation of 25(OH)D_3_ during fasting was impaired in the current study. If fasting induced the formation of 25(OH)D_3_ metabolites, such as 24,25(OH)_2_D_3_, 3-*epi*-25(OH)D_3_, and 1,25(OH)_2_D_3_ (not significantly), and was additionally accompanied by a decrease in its synthesis, a significant decrease in serum 25(OH)D_3_ levels could be expected; however, that was not the case. Interestingly, fasting upregulates the synthesis of CYP24A1, an enzyme that catalyzes the 24-hydroxylation of 25(OH)D_3_ and 1,25(OH)_2_D_3_. The data presented in the current study support this notion, as a significant increase in serum 24,25(OH)_2_D_3_ levels was observed after fasting. The 24-hydroxylation is considered to be the main inactivating step limiting the biological activity of 1,25(OH)_2_D_3_. However, this might not be the only biological effect of these reactions. One product of CYP24A1 is 24,25(OH)_2_D_3_, which performs several biological functions. Specifically, it protects against 1,25(OH)_2_D_3_ toxicity, is anti-inflammatory, stimulates bone healing, and can exert antioxidant effects [17,18]. Hence, it cannot be excluded that the increase in 24,25(OH)_2_D_3_ levels during fasting is an adaptive response to the oxidative challenge induced by fasting. 

Vitamin D has some anti-inflammatory effects, which have been related to an increased expression of IL-10 [7]. Consequently, we initially assumed that changes in vitamin D metabolism could be associated with changes in IL-10; however, this was not confirmed. Another metabolite of 25(OH)D_3_ whose concentration increased after 8-d fasting was 3-*epi*-25(OH)D_3_. C-3 epimerization is a common metabolic pathway and 3-*epi*-25(OH)D_3_ is considered to be an inactive form of vitamin D [19]. However, some studies indicate that it does have some biological functions. For example, 3-*epi*-25(OH)D_3_ can give rise to 3-*epi*-1,25(OH)D_3_, which suppresses parathormone secretion without inducing changes in blood calcium levels, unlike 1,25-(OH)D_3_ [5]. It also stimulates surfactant phospholipid synthesis in pulmonary cells [20]. Further, it has been suggested that epimerization may be the first line of defense of the body against high levels of 25(OH)D_3_, as they rise promptly after supplementation with high doses of vitamin D [14]. The data presented herein suggest that 3-*epi*-1,25(OH)D_3_ can also play another role, which needs to be determined. We observed an increase in 3-*epi*-25(OH)D_3_ levels after fasting in subjects whose 25(OH)D_3_ status was below 20 ng/mL, indicating vitamin D insufficiency [21]. Hence, it cannot be assumed that increased 3-*epi*-25(OH)D_3_ levels in individuals with a low vitamin D status serve to protect from vitamin D toxicity. In a population study of over 8000 individuals, only 7.7% of subjects had measurable 3-*epi*-25(OH)D_3_ levels. The authors concluded that quantification of 3-*epi*-25(OH)D_3_ levels does not significantly influence the clinical interpretation of vitamin D levels [22]. In the current study, 3-*epi*-25(OH)D_3_ was detected in all subjects. This apparent discrepancy between the current and former studies stems from the limits of detection of the methods used, which were >5 nmol/L in the study of Karefylakis et al. [21] and 0.5 nmol/L in the current study. The current study and our previous studies, in which 3-*epi*-25(OH)D_3_ was confirmed in all studied subjects, suggest that this vitamin D metabolite can play an important biological role, which is worth future study. Conversely, in the current study, the increase in 24,25(OH)_2_D_3_ and 1,25(OH)_2_D_3_ levels (not significant) after fasting was observed only in subjects with 25(OH)D_3_ levels that were higher and lower than 20 ng/mL, respectively. It is important to note that baseline levels of 3-*epi*-25(OH)D_3_, 24,25(OH)_2_D_3_, and 1,25(OH)_2_D_3_ were higher in subjects with 25(OH)_2_D_3_ levels above 20 ng/mL. These observations clearly indicate that fasting induces changes in vitamin D metabolism, which are partially dependent on serum 25(OH)D_3_ levels.

The effects of exercise combined with fasting on vitamin D3 metabolism constitute another interesting aspect of the current study. After overnight fasting, the exercise temporarily (1 h after the exercise) increased serum 3-*epi*-25(OH)D levels, with no changes in other metabolite levels. Similarly, no changes in 25(OH)D_3_ levels after a maximal incremental exercise (similar to the one performed in the current study) was observed in young men and women in one study [23]. Conversely, in another study, 30-min cycling exercise increased 25(OH)D_3_ levels, with no effect on 1,25(OH)_2_D_3_ levels [10]. This discrepancy could be explained by a difference in the exercise test and duration of exercise. In the current study, the subjects performed the progressive test for just 16–18 min. By contrast, a similar exercise test performed after 8 d of fasting significantly increased serum 24,25(OH)_2_D_3_, 3-*epi*-25(OH)D, 25(OH)D_3_, and 25(OH)_2_D_2_ levels. 

The effects of exercise on vitamin D metabolite levels after fasting were dependent on basal 25(OH)D_3_ levels. Increased serum 24,25(OH)_2_D_3_ levels were only observed in subjects with basal 25(OH)D_3_ levels above 20 ng/mL. It can thus be concluded that exercise-induced changes in vitamin D metabolite levels are dynamic. The mechanism of this phenomenon remains to be determined. The current study supports the findings of Sun et al. (2017), who observed an increase in serum 25(OH)D_3_ levels 30 min after exercise, followed by their decrease and another increase after 3 h and 24 h [10]. In another study, 25(OH)D_3_ levels remained elevated 24 h after an ultramarathon [14]. 

## 5. Conclusions

In conclusion, this study demonstrates that fasting and exercise significantly modulate vitamin D metabolism. As a caveat, the detection limit of the method used for the quantification of vitamin D metabolites in the current study is much lower than that of methods used in some previously published studies. Increased serum levels of 24,25(OH)_2_D_3_ and 3-*epi*-25(OH)D_3_ after fasting and exercise indicate that these metabolites play some important biological roles and are not simply end products of vitamin degradation. Further, formation of 24,25(OH)_2_D_3_ and 1,25(OH)_2_D_3_ after fasting was dependent on the basal levels of 25(OH)D_3_. Nonetheless, serum levels of all vitamin D metabolites analyzed herein were elevated in individuals with higher basal levels of 25(OH)D_3_. These data indicate that in addition to some food components, fasting and exercise can be included as factors that can modify vitamin D metabolism.

## Figures and Tables

**Figure 1 nutrients-13-01963-f001:**
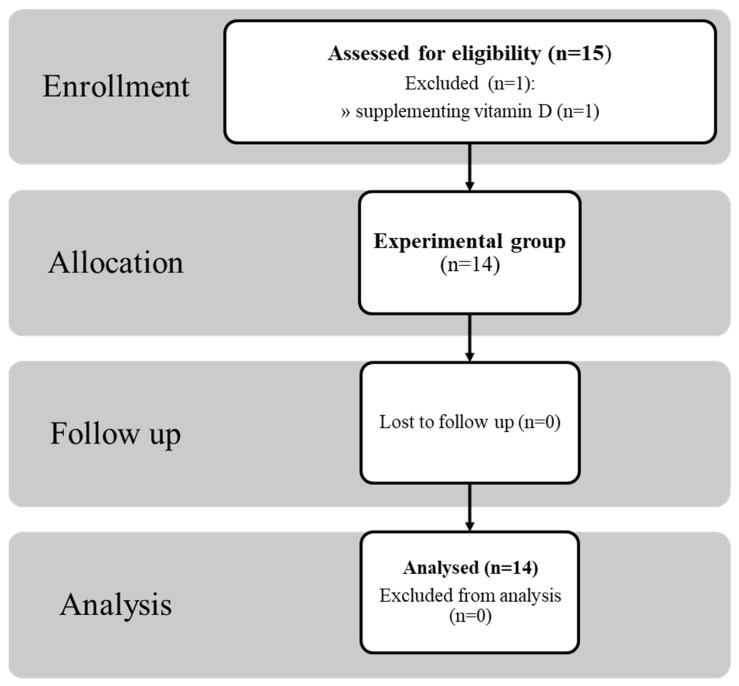
Flow diagram of participant progress through the phases of the study.

**Figure 2 nutrients-13-01963-f002:**
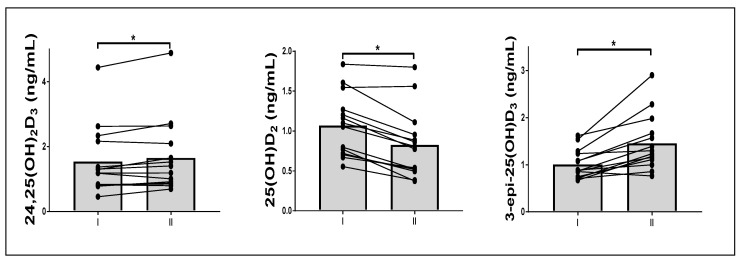
Starvation-induced changes in vitamin D metabolism. I, after overnight fasting; II, after 8 d of fasting. Shapes indicate the means (* *p* < 0.05, paired tests).

**Figure 3 nutrients-13-01963-f003:**
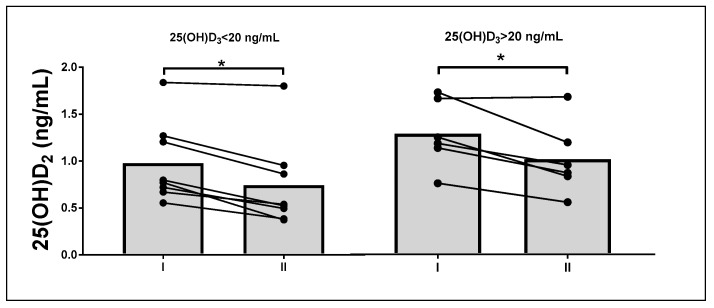
Effects of 8 d of fasting on serum levels of 25(OH)D_2_ in subgroups with different baseline 25(OH)D_3_ levels (by repeated-measures analyses of variance with post-hoc tests, * *p* < 0.05).

**Table 1 nutrients-13-01963-t001:** Anthropometric and physiological characteristics of the study participants.

	I	II	Change	*p*
BW (kg)	79.9 ± 10.98	74.8 ± 11.0	−5.6 ± 1.4	***<0.01***
Fat (kg)	15.8 ± 5.5	13.9 ± 5.7	−2.0 ± 0.6	**<0.01**
FFM (kg)	64.1 ± 5.8	60.7 ± 6.1	−3.8 ± 1.3	**<0.01**
BMI (kg/m^2^)	25.1 ± 3.3	23.4 ± 3.4	−1.8 ± 0.5	***<0.01***
VO_2_ max (ml/min/kg)	47.0 ± 12.0	40.0 ± 7.3	−7.1 ± 7.8	**<0.01**
Load (W)	212.1 ± 38.1	180.0 ± 36.7	−32.3 ± 25.9	**<0.01**

Values are means ± SD. I, after overnight fasting; II, after 8 d of fasting; BW, body weight; Fat, body fat; FFM, fat-free mass; BMI, body mass index; VO_2_ max, maximal oxygen consumption; Load, the amount of work done by the participant. Paired *t*-test (normal font) or the Wilcoxon test (italic) were used to evaluate the differences, significant changes are in bold, *p* < 0.05.

**Table 2 nutrients-13-01963-t002:** Changes in vitamin D metabolite levels after exercise before and after fasting.

	I	I h	*p* (I vs. I h)	II	II h	*p* (II vs. II h)
25(OH)D_3_ (ng/mL)	21.7 ± 6.9	22.3 ± 8.1	0.17	21.0 ± 7.9	23.2 ± 9.1	**0.00**
24,25(OH)_2_D_3_ (ng/mL)	1.5 ± 1.1	1.6 ± 1.1	*0.36*	1.7 ± 1.2	1.9 ± 1.4	***0.02***
25(OH)D_2_ (ng/mL)	1.1 ± 0.4	1.1 ± 0.5	*0.19*	0.8 ± 0.3	0.8 ± 0.3	**0.01**
3-*epi*-25(OH)D_3_ (ng/mL)	1.0 ± 0.3	1.1 ± 0.3	0.00	1.5 ± 0.6	1.6 ± 0.7	***0.00***
1,25(OH)_2_D_3_ (pg/mL)	64.1 ± 23.4	69.7 ± 27.9	*0.34*	72.6 ± 23.4	74.5 ± 28.7	*0.76*

Values are means ± SD. I, before fasting; I h, before fasting, 1 h after exercise; II, after 8 d of fasting; II h, after 8 d of fasting, 1 h after exercise. Paired *t*-test (normal font) or the Wilcoxon test (italic) were used to evaluate the differences, significant differences are in bold, *p* < 0.05.

**Table 3 nutrients-13-01963-t003:** Differences in vitamin D metabolite levels in subgroups with different baseline 25(OH)D_3_ levels.

	Baseline 25(OH)D_3_		*p*
	<20 ng/mL	>20 ng/mL	
25(OH)D_3_ (ng/mL)	17.4 ± 2.5	27.5 ± 6.5	**<0.01**
24,25(OH)_2_D_3_ (ng/mL)	0.9 ± 0.3	2.4 ± 1.2	**0.01**
25(OH)D_2_ (ng/mL)	1.0 ± 0.4	1.2 ± 0.3	0.33
3-epi-25(OH)D_3_ (ng/mL)	0.8 ± 0.2	1.3 ± 0.3	***0.01***
1,25(OH)_2_D_3_ (pg/mL)	48.8 ± 16.6	82.5 ± 15.8	**0.01**

Values are means ± SD, unpaired *t*-test (normal font) or the Wilcoxon test (italic) were used to evaluate the differences.

**Table 4 nutrients-13-01963-t004:** Changes in vitamin D metabolite levels induced by a single exercise after 8 d of fasting depending on the baseline 25(OH)D_3_ levels.

	25(OH)D_3_ < 20 ng/mL		25(OH)D_3_ > 20 ng/mL		rANOVA	
	II	II h	II	II h	*p*	$\eta_{p}^{2}$
25(OH)D_3_ (ng/mL)	15.57 ± 2.55	17.07 ± 2.86	27.27 ± 7.41	30.45 ± 8.42 *	0.04	0.31
24,25(OH)_2_D_3_ (ng/mL)	0.95 ± 0.27	1.02 ± 0.17	2.56 ± 1.25	2.93 ± 1.6 *	0.05	0.24
25(OH)D_2_ (ng/mL)	0.59 ± 0.23	0.65 ± 0.23	0.94 ± 0.36	1.01 ± 0.34	0.78	0.01
3-epi-25(OH)D_3_ (ng/mL)	1.08 ± 0.22	1.16 ± 0.21	1.93 ± 0.61	2.08 ± 0.68	0.10	0.07
1,25(OH)_2_D_3_ (pg/mL)	61.87 ± 22.89	59.32 ± 28.83	85.42 ± 18.21	92.81 ± 16.12	0.45	0.07

Values are means ± SD. II, after 8 d of fasting; II h, after 8 d of fasting, 1 h after exercise. *—statistically significant differences (by repeated-measures analyses of variance with post-hoc tests), $\eta_{p}^{2}$—the effect size.

## Data Availability

All the data are available from the first author under reasonable request.

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
