# Peer review of "Fasting and Exercise Induce Changes in Serum Vitamin D Metabolites in Healthy Men"

_nutrients, 2021, doi:10.3390/nu13061963_

Round 1
Reviewer 1 Report
This is an important study on the effects of exercise and fasting on serum vitamin D metabolites. There is very little information if any on the role of fasting and excersie on short term changes in vitamin D metabolites. Measurement if different vitamin D metabolites is a strength of this study. A number of areas need clarification to improve this manuscript
- More details on the fasting protocol needs to be added. How many calories were allowed each day during the fast? Any other intake data? Was this a complete starvation? What adverse effects were reported by the participants? More details should be added in this section
- In the flow chart of participants- How much vit D was taken by the person who was supplementing vit D?
- Table 1: Expand abbreviations in the footnote. What is I and II ? This is not clear? What is Load? How was this calculated?
- The lack of a control group is a limitation. Please discuss this in the methods along with alternate approaches
Author Response
Response to Reviewer 1.
At first, we'd like to thank you very much for your valuable comments and review of our manuscript.
- The Reviewer wrote: “More details on the fasting protocol needs to be added. How many calories were allowed each day during the fast? Any other intake data? Was this a complete starvation? What adverse effects were reported by the participants? More details should be added in this section”
Response: Thank you for the comments some changes in the text has been introduced. During the fasting period, the subjects only consumed mineral water, which contained an average amount of ions. No any food or drink containing calories were allowed. During the study no, adverse effects of fasting has been recorded. For safety reasons, the men remained under medical care for 3 days before the
study, during the 8-day of fasting and 3 days after its completion.
- The Reviewer wrote: “In the flow chart of participants- How much vit D was taken by the person who was supplementing vit D?”
Response: One subject was excluded from the analysis as he used vitamin D supplements (around 10 000 IU per day) for several weeks before the study.
- The Reviewer wrote: ”Table 1: Expand abbreviations in the footnote. What is I and II ? This is not clear? What is Load? How was this calculated?”
Response: Footnote has been corrected I, after overnight fasting; II, after 8 d of fasting. The initial load (60 W) was increased gradually every 3 min by 30 W.
As the test was until exhaustion the subjects differ between themselves by maximal load no other calculation has been made.
- The Reviewer wrote: “The lack of a control group is a limitation. Please discuss this in the methods along with alternate approaches”.
Response: It has been introduced in methods section “The study did not include control group as the effects of exercise has been compared before and after fasting in the same subject thus they were control for themselves”.
Thank you very much again for all comments, which raised the level of our manuscript.
Sincerely, Jedrzej Antosiewicz
Reviewer 2 Report
In the current manuscript entitled " Fasting and exercise induce changes in serum vitamin D metabolites in healthy men ", Żychowska et al studied the relationship between vitamin D metabolites, fasting and yoga. After 8 day of fating, volunteers have higher levels of 24,25(OH)2D3 and 3-epi-25(OH)D3 and lower 25(OH)D2. Furthermore, exercise exerts increases in 25(OH)D3, 24,25(OH)2D3, and 3-epi-25(OH)D3, and a reduction in 25(OH)D2. Increase in 24,25(OH)2D3 and 3-epi-25(OH)D3 levels implies that fasting alters vitamin D distribution.
- There are a number of places in the abstract that do not write correctly.
- In the introduction section, authors should add the review of vitamin D and inflammation.
- In the method section, please provide the measurement of b-hydroxybutyrate.
- Authors must identify the actual tests included in the statistical analysis section. Which statistical tests did authors use?
- Authors should discuss the potency of C3-epimerization of vitamin D.
Author Response
Response to Reviewer 2
At first, we'd like to thank you very much for your valuable comments and review of our manuscript.
- The Reviewer wrote: “There are a number of places in the abstract that do not write correctly”.
Response: The abstract has been check for accuracy.
- The Reviewer wrote: “In the introduction section, authors should add the review of vitamin D and inflammation”.
Response: Thank you for the comment it has been introduced
- The Reviewer wrote: “In the method section, please provide the measurement of b-hydroxybutyrate.”
Response: b-Hydroxybutyrate levels were determined using a commercial kit (RANBUT diagnostic kit, RANDOX, Crumlin, UK).
- The Reviewer wrote: “Authors must identify the actual tests included in the statistical analysis section. Which statistical tests did authors use?”
Response: It has been corrected
- The Reviewer wrote: “Authors should discuss the potency of C3-epimerization of vitamin D.”
Response: Thank you for the comment. We fully agree with the Reviewer however to the best of our knowledge a data about C3-epimer are very scarce and all we know about the physiological function of it is included in the discussion.
Thank you very much again for all comments, which raised the level of our manuscript.
Sincerely, Jedrzej Antosiewicz
Round 2
Reviewer 1 Report
All previous comments are addressed
Author Response
Thank you